# The visual side of knowledge: the role of images in Wikipedia and Wikidata

This proposal addresses a critical yet underexplored aspect of Wikimedia projects: the visual representation of knowledge. While extensive research has examined the textual content and its technical and social dimensions, the study of images as representations of knowledge remains significantly less developed (Rama et al., 2022; Viegas, 2007). Often treated as supplementary information, images however play a crucial role in conveying information and shaping the representation and understanding of concepts (Menking et al., 2018). This is true not only for photographs and digitized artifacts, such as portraits, but also for composited images like maps, charts, and diagrams (Kosminsky et al., 2022).

This study aims to bridge this gap by identifying the most viewed images on Wikimedia Commons and analyzing their usage across Wikipedia and Wikidata.

Key questions include: Which images are used to represent specific concepts? Which images are designated as the primary representation of Wikidata items through the image (P18) property (Meta-Wiki, 2021)? And which images have the greatest visual impact across Wikimedia projects?

Methodologically, the study is structured as follows: The first step involves identifying the "most seen" images. Unlike Wikimedia pages, it is challenging to determine precisely when an image is "viewed," as no publicly accessible data directly addresses this. Instead, the study uses a proxy metric: mediacounts, which measure the number of times an image is delivered to a client computer, either as a thumbnail or in high resolution. While this metric does not fully account for technical factors like preloading by client browsers, it serves as a reliable approximation.

By collecting and aggregating mediacount data over the past year, the study compiles a list of the 10,000 most requested images, both as thumbnails and in high-quality formats. Images are then classified based on their content (e.g., portraits, insignia, diagrams, maps).

Finally, each image's usage on Wikidata is analyzed, focusing on which properties link them.

The study's results highlight how the most viewed images play an identity-defining role, providing unique representations for various concepts. This role is highly correlated with their usage on Wikidata. Images such as portraits, flags, insignia, and chemical representations feature prominently among the most viewed. These findings underscore the critical role of Wikidata in elevating certain images as primary representations of items, often facilitated through their automatic inclusion in templates.

Kosminsky, D., Walny, J., Vermeulen, J., Knudsen, S., Willett, W., & Carpendale, S. (2022). Belief at first sight: Data visualization and the rationalization of seeing. *Information Design Journal*, 43–55. https://doi.org/10.1075/idj.25.1.04kos

Menking, A., Rangarajan, V., & Gilbert, M. (2018). "Sharing small pieces of the world": Increasing and broadening participation in Wikimedia Commons. *Proceedings of the 14th International Symposium on Open Collaboration*, 1–12. https://doi.org/10.1145/3233391.3233537

Meta-Wiki. (2021). *Research:Recommending Images to Wikidata Items—Meta*. Meta.Wikimedia.Org. https://meta.wikimedia.org/w/index.php?title=Research:Recommending_Images_to_Wikidata_Items&oldid=21588160

Rama, D., Piccardi, T., Redi, M., & Schifanella, R. (2022). A large scale study of reader interactions with images on Wikipedia. *EPJ Data Science*, *11*(1), 1. https://doi.org/10.1140/epjds/s13688-021-00312-8

Viegas, F. (2007). The Visual Side of Wikipedia. *2007 40th Annual Hawaii International Conference on System Sciences (HICSS'07)*, 85–85. https://doi.org/10.1109/HICSS.2007.559
