# OpenReview forum: "The visual side of knowledge: the role of images in Wikipedia and Wikidata"
_wikimedia.it/Wikidata_and_Research/2025/Conference — Submitted to WD&R_

### Official Review · ~Daniel_Mietchen1 · 2025-01-19
**A useful attempt at analyzing image use on Commons, Wikipedia and Wikidata**

**Originality:** 5
**Impact:** 4
**Confidence:** 4

**Review:**

By focusing on images, the authors have commendably chosen an important gap in research about Wikimedia projects. The outlined methodology seems appropriate to the matter. In terms of relative coverage, there is room for expanding on both the Wikidata aspects and on the research part. Some examples of issues in this realm would be
(i) does image usage on Wikidata differ between research-related topics and other topics,
(ii) what differences can be observed by research fields,
(iii) do images from research-related sources get used differently than others,
(iv) which aspects of the research process or the research landscape get most of the image attention, and which get none,
(v) what trends can be observed in terms of languages, time or geography,
(vi) besides these 10k most requested images, is there something to be said about the long tail,
(vii) besides the properties having images as objects, what about properties of items that have images?
In terms of impact on the Wikimedia or academic communities, I can imagine this to be substantial, but the submission does not provide much detail on that.

**Compliance:**

4

**Notes:**

For the "Compliance" evaluation, I used "Compliance of the proposal with the conference themes" (as stated at https://w.wiki/CmYB ), rather than "Compliance with Wikimedia guidelines and standards" (as stated in the evaluation form).

**Scientific Quality:**

5

---

### Official Review · ~Alessio_Melandri1 · 2025-01-21
**Unveiling the role of images in Wikimedia knowledge representation**

**Originality:** 4
**Impact:** 5
**Confidence:** 5

**Review:**

This proposal addresses a compelling and underexplored dimension of Wikimedia projects: the role of images as primary vehicles of knowledge representation. The study’s focus on mediacounts as a proxy for image visibility is innovative and pragmatic, offering valuable insights into the visual impact of Wikimedia Commons content. By analyzing the interplay between image prominence and usage in Wikidata, the research sheds light on the identity-defining role of visual elements in open knowledge ecosystems. While the methodology is robust, it may benefit from broader contextualization of mediacount limitations. The findings hold promise for influencing how visual resources are prioritized and integrated across Wikimedia platforms.

**Compliance:**

5

**Scientific Quality:**

5

---

### Decision · Program_Chairs · 2025-02-05

**Decision:**

Withdrawn

**Comment:**

== Ritirato dall'autore ==


Dear Author,
thank you very much for your proposal. We regret to inform you that your proposal was not selected among the papers.

Even if not selected as paper, we consider your proposal relevant and interesting and we would like to propose you to prepare instead a lightening talk (if you - or another member of your team - can participate in presence at the conference) or a poster (which can be exhibited even if you will not attend the conference).

It would be a pleasure to learn more about your work through a lightening talk or a poster.
Thank you for submitting a proposal and please let us know if you like the idea of converting it into a lightening talk or a poster and which format you prefer.

Regards,
The scientific committee of the conference Wikidata and Research